# Video-observed therapy (VOT) vs directly observed therapy (DOT) for tuberculosis treatment: A systematic review on adherence, cost of treatment observation, time spent observing treatment and patient satisfaction

**Thiago Areas Lisboa Netto**[1]*, **Bruna Dellatorre Diniz**[2], **Peter Odutola**[3], **Clara Rocha Dantas**[4], **Maria Carolina Fonseca Loureiro Caldeira de Freitas**[5], **Philip Michael Hefford**[6], **Taniela Marli Bes**[7]

1 Evandro Chagas National Institute of Infectious Diseases (Fiocruz), Rio de Janeiro, Brazil, 2 Western Michigan University, Kalamazoo, Michigan, United States of America, 3 Harvard University, Cambridge, Massachusetts, United States of America, 4 Universidad de Buenos Aires, Ciudad Autonoma de Buenos Aires, Argentina, 5 University of Brasilia, Brasilia, Brazil, 6 University of Sussex, Brighton, United Kingdom, 7 MetroWest Medical Center, Tufts University, Massachussets, United States of America

* thiagonetto68@gmail.com

## Abstract

### Introduction

Tuberculosis (TB) treatment demands strict adherence to multidrug regimens. Directly Observed Therapy (DOT) poses challenges, especially regarding adherence. With the popularization of smartphones, Video-Observed Therapy (VOT) has emerged as a promising alternative, allowing healthcare providers to remotely supervise patients taking their medications via video calls.

### Objectives

This systematic review critically assesses VOT's effectiveness compared to DOT, focusing on adherence, treatment costs, time spent supervising treatment, and patient satisfaction, aiming to optimize TB supervision methods worldwide.

### Methods

Only studies that met the following criteria were eligible for inclusion in the systematic review: randomized trials; studies that compared VOT to DOT; studies involving patients diagnosed with pulmonary or extrapulmonary tuberculosis; studies that reported any of the desired outcomes; full-text articles available for review; and studies conducted in the English language. We excluded studies with the following attributes: studies that lacked a control group; case series or case reports; and previous systematic reviews.

The search engines and databases MEDLINE, Embase, and Cochrane were used to find studies comparing Video-Observed Therapy (VOT) to Directly Observed Therapy (DOT).

**Data Availability Statement:** All relevant data are in the manuscript and its supporting information files.

**Funding:** The author(s) received no specific funding for this work.

**Competing interests:** The authors have declared that no competing interests exist.

The following search phrases were used to look for papers that contained them in their title or abstract: ("Electronic Directly Observed Therapy" OR "Video-observed therapy" OR "Telemedicine" OR "Wirelessly observed therapy" OR "Smartphone-enabled video-observed") AND ("TUBERCULOSIS").

## Results

A systematic review of the literature revealed the following findings: in all Randomized Controlled Trials (RCTs), video-observed therapy (VOT) demonstrated non-inferiority in terms of treatment adherence compared to traditional directly observed therapy (DOT); VOT reduced costs where these outcomes were assessed in the RCTs; the use of VOT reduced the amount of time healthcare professionals spent supervising treatment in RCTs evaluating this aspect; VOT contributed to higher treatment satisfaction in RCTs where this outcome was measured.

## Conclusion

In this systematic review we emphasize the importance of Video-Observed Therapy (VOT) in the digital age for patients that have access to internet. Our findings show that VOT is comparable to DOT in terms of treatment adherence, but it is also cost-effective, improves patient satisfaction and takes less time for healthcare professionals to supervise.

### Author summary

Our study compares two methods of monitoring tuberculosis treatment: Video-observed therapy (VOT) and directly observed therapy (DOT). Tuberculosis is an infectious disease that requires strict adherence to treatment, and traditionally, healthcare workers physically observe patients taking their medication (DOT). However, this can be time-consuming and costly. VOT, where patients are observed via video, has emerged as a possible alternative.

In this systematic review, we evaluated how VOT and DOT compare in terms of patient adherence, the costs associated with treatment observation, the time required to observe treatment, and patient satisfaction. We found that VOT generally leads to similar or better adherence compared to DOT, while also being more cost-effective and time-efficient. Patients also reported higher satisfaction with VOT, likely due to its convenience and flexibility. Our findings suggest that VOT could be a valuable tool in improving tuberculosis treatment outcomes, especially in resource-limited settings.

## 1. Introduction

Tuberculosis (TB) remains a significant global health challenge and its treatment involves a multidrug regimen taken over several months, requiring strict adherence to ensure successful outcomes and prevent the development of drug-resistant strains [1, 2]. Directly Observed Therapy (DOT) has long been considered the gold standard for TB treatment supervision. It consists of a healthcare provider or trained observer watching the patient swallowing their medication in person to ensure compliance. However, the conventional DOT approach poses logistical challenges, especially in low and middle-income countries (LMICs), where the

majority of TB cases occur, including substantial time and resource commitments from healthcare providers and patients [3]. With advancements in digital technologies, Video-Observed Therapy (VOT) has emerged as a promising alternative, allowing healthcare providers to observe patients taking their medications via video calls remotely [4].

In this systematic review, we compare the effectiveness of Video-Observed Therapy with Directly Observed Therapy in terms of adherence rates, the cost of treatment observation, staff time spent on observation, and patient satisfaction. Using existing evidence, this review provides valuable insights into the pros and cons of both approaches, suggesting recommendations that are evidence-based for optimizing TB treatment supervision methods in diverse healthcare settings.

## 2. Methods

### 2.1 Eligibility criteria

Only studies that met the following criteria were eligible for inclusion in the systematic review: randomized trials; studies that compared Video-Observed Therapy (VOT) to Directly Observed Therapy (DOT); studies involving patients diagnosed with pulmonary or extrapulmonary tuberculosis; studies that reported any of the desired outcomes; full-text articles available for review; and studies conducted in the English language. We excluded studies with the following attributes: studies that lacked a control group; case series or case reports; and previous systematic reviews.

### 2.2 Search strategy and screening

A comprehensive database search was conducted in accordance with the Preferred Reporting Items for Systematic Reviews and Meta-Analyses (PRISMA) guidelines. A filled-out PRISMA checklist is provided as an appendix to this manuscript. The database search, carried out in July 2023, utilized MEDLINE, Embase, and Cochrane to find studies comparing VOT with DOT. The search terms used included: ("Electronic Directly Observed Therapy" OR "Video-observed therapy" OR "Telemedicine" OR "Wirelessly observed therapy" OR "Smartphone-enabled video-observed") AND ("TUBERCULOSIS"). Only English-language texts were included. Two authors independently reviewed the identified manuscripts for potential inclusion, evidence grading, and data extraction. A third author was responsible for mediating any differences between the identified data.

Directly Observed Therapy (DOT) was defined differently across studies. In some instances, DOT involved in-person delivery of medication to the patient's home, while in other cases, patients were required to travel to a specific location to receive their medication. These variations in DOT implementation likely had a substantial effect on adherence and patient preference.

### 2.3 Endpoints of the systematic review

Adherence to therapy was the main outcome of interest. Cost of treatment observation, staff time spent observing treatment and patient satisfaction were secondary objectives.

### 2.4 Data extraction

The inclusion and data extraction of each study was evaluated separately by two authors. In addition to authorship, country of study, study duration, sample size, age and gender, detailed in Table 1, the following information was pertinent to this systematic review: adherence to

**Table 1. Baseline characteristics of the studies.**

| Author, year and reference | Study country | Duration of study | Sample size (DOT / VOT) | Gender (men/ women) | Age (years) | Primary outcome |
|---|---|---|---|---|---|---|
| Burzynski, 2022 [9] | USA | July/2017—Oct/2019 | 103 / 113 | 140 / 76 | 16–86 | Differences in the % of medication doses between DOT and VOT |
| Ravenscroft, 2020 [8] | Moldova | Jan/2016—Nov/2017 | 99 / 98 | 97 / 81 | ≥18 | Number of days over 2 weeks that a patient was not observed taking medication or did not attend appointments |
| Guo, 2019 [7] | China | Jan/2018—Dec/2018 | 202 / 203 | 284 / 121 | 18–89 | % of patients that achieve good treatment outcome: cured + treatment completed (WHO criteria) |
| Story, 2019 [6] | UK | Sep/2014—Oct/2016 | 114 /112 | 165 / 51 | 16–34: 123 patients 35–54: 80 patients ≥ 55: 21 patients | % of the patients that complete 80% or more scheduled treatment observations over the first 2 months |

% = percentage; WHO = World Health Organization

therapy, cost of treatment observation, time spent observing treatment and patient satisfaction.

One author extracted the data from the included studies, and the other author double-checked the data. Discussion amongst the writers was used to settle disagreements. The collected information was stored in an Excel spreadsheet.

Due to substantial heterogeneity in the reporting of data across the included studies, including how adherence was measured, the cost of treatment, time spent on treatment, and treatment satisfaction, these variables are detailed in Table 2. This table summarizes the different approaches and results reported by the included studies to provide a comprehensive view of the observed variations and facilitate comparison across the different contexts and methods used.

### 2.5 Quality assessment

Using the Cochrane risk-of-bias tool for measuring bias in randomized trials, the included randomized controlled trials (RCTs) were evaluated for quality [5]. Accordingly, studies are rated in five categories—selection, performance, detection, attrition, and reporting—as high, low, or ambiguous.

### 2.6 Data analysis

After reading the papers in full, the following data were compiled and analyzed in Tables 1 and 2.

## 3. Results

A total of 910 studies were identified, of which 891 were excluded either for being unrelated to the topic of discussion or for being duplicates. Of the remaining 19 studies, 4 met the inclusion criteria and were included in the analysis. In total, 1,044 patients were randomized across these selected studies. These data are summarized in Fig 1. The baseline characteristics of the studies are shown in Table 1, and the summary of the findings from each analysis is detailed in Table 2.

### 3.1 Characteristics of the included studies

The study periods ranged from 2014 to 2019, and they were conducted in Moldova, England, China, and the USA. Each study enrolled both male and female participants, with

**Table 2. Review findings.**

| Author, year and reference | Study Findings | | | |
|---|---|---|---|---|
| | Adherence | Cost of treatment observation per patient (average cost) | Mean time spent observing treatment per patient | Patient satisfaction |
| Burzynski, 2022 [9] | % of completed doses:<br>DOT: 87.2% (95% CI, 84.6%-89.9%)<br>VOT: 89.9% (95% CI, 87.5%-92.1%) | NA | NA | Among participants completing the crossover period, 84% preferred VOT, while 5.6% preferred DOT. |
| Ravenscroft, 2020 [8] | Days of missed observation (mean/2 week period):<br>DOT 5.24 vs VOT: 1.29<br>% of patients with ≥ 80% of adhrence:<br>DOT: 19.5% vs VOT: 74.5%; | DOT: 39.13 USD<br>VOT: 10.5 USD | Total time:<br>DOT: 81 h<br>VOT: 23 h | Patient satisfaction rating on 5-point scale (mean):<br>DOT: 4.2 vs VOT: 4.9<br>Cumulative log-odds increment (VOT/DOT): 3.29 (95% CI 1.66–4.92); |
| Story, 2019 [6] | % of patients meeting ≥ 80% of scheduled observations:<br>DOT: 35/114 patients (31%)<br>VOT: 78/122 patients (70%)<br>OR: 5·48 (95% CI: 3·10–9·68) | VOT (daily): 2072,86 USD<br>†DOT: 7182,57 USD | Time per week:<br>VOT: 1.8 min. (SD 2.2)<br>DOT: 29 min. (SD 48); | Participants' responses to the statement "I am satisfied with the way my treatment is observed":<br>DOT:<br>- Strongly agree: 30 (52.6%)<br>- Agree: 24 (42.1%)<br>- Neither agree nor disagree: 1 (1.8%)<br>- Disagree: 2 (3.5%)<br>- Strongly disagree: 0 (0%)<br>VOT:<br>- Strongly agree: 29 (44.6%)<br>- Agree: 30 (46.2%)<br>- Neither agree nor disagree: 1 (1.5%)<br>- Disagree: 0 (0%)<br>- Strongly disagree: 3 (4.6%) |
| Guo, 2019 [7] | Participants' treatment completion:<br>DOT: 94.6%; vs VOT: 96.1% | ‡DOT: 10.1 USD (SD 49.7)<br>VOT: 4.83 USD (SD 3.8) | Time per dose:<br>DOT: 44.1 min. (SD 32.7)<br>VOT: 16.5 min. (SD 12.1) | % of patients finding supervising method convenient and comfortable:<br>DOT: 111 (56%) vs VOT: 191 (96%)<br>% of patients reporting that supervising method assisted in dose reminders:<br>DOT: 171 (86,7%) vs VOT: 185 (93%) |

OR: odds-ratio; SD: standard deviation; NA: No data available

USD: US dollars; (conversion to USD was made by the authors)

†: DOT was made 5 times per week

‡: includes the cost of transportation

% = percentage; / = divided by; 95% CI = 95% confidence interval

a greater proportion of men compared to women (686 men versus 329 women) and with the age range encompassing individuals aged 16 and above. All studies were randomized controlled trials.

### 3.2 Review findings

All studies demonstrated either non-inferiority or increased adherence with video observed treatment, the primary outcome measure. Additionally, three out of four studies indicated a significant reduction in average staff time per dose for patients assigned to video observed treatment. Furthermore, the costs associated with providing direct observed treatment were notably higher compared to video observed treatment. In terms of participant experience, two studies found that patients preferred video observed treatment over direct observation.

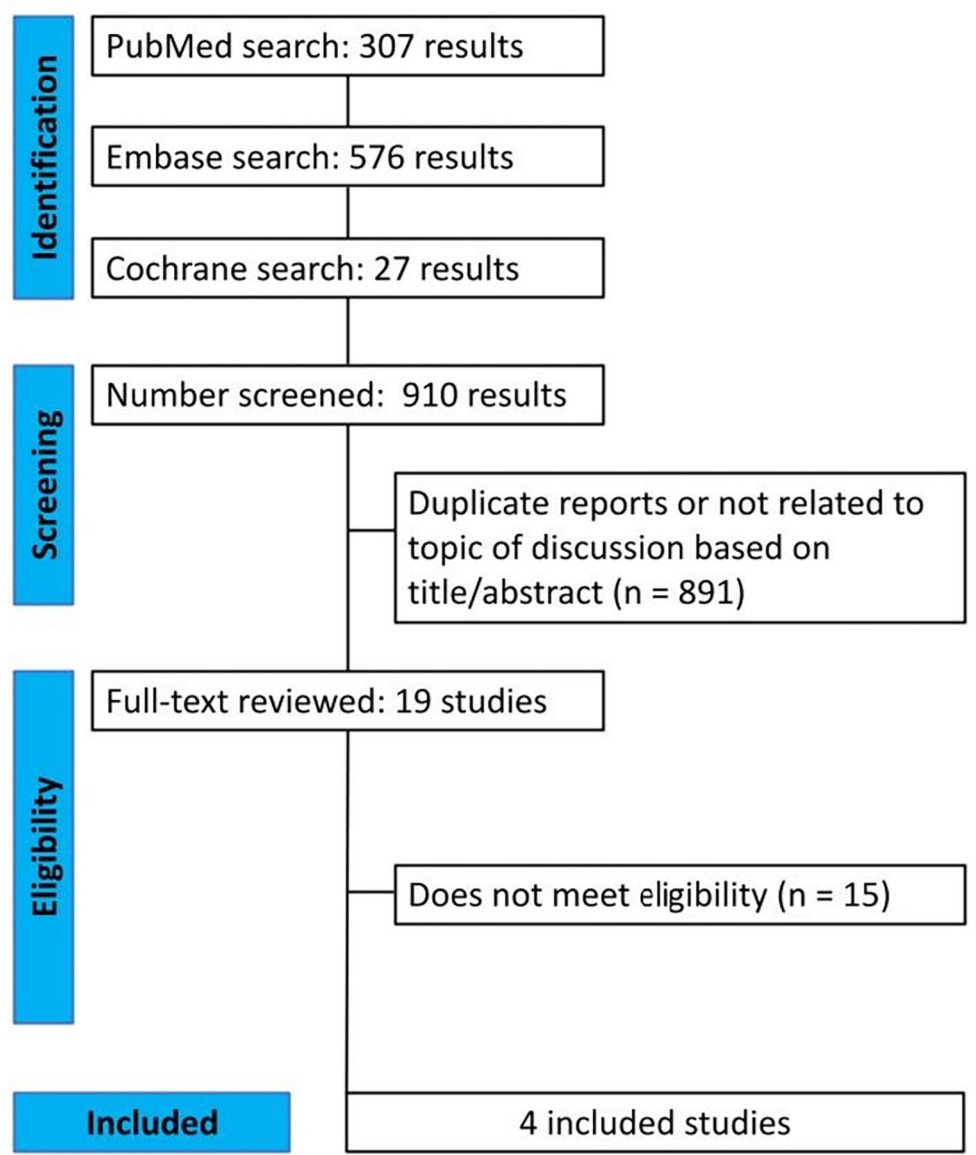

**Fig 1. PRISMA flow diagram of study screening and selection.**

### 3.3 Quality assessment

The risk of bias in each study using the Cochrane risk-of-bias tool is detailed in Table 3. Over-all, the studies exhibited a low risk of selection bias. Performance bias was consistently high

**Table 3. Quality assessment using the Cochrane risk-of-bias tool.**

| Study | Selection Bias | Performance Bias | Detection Bias | Attrition Bias | Reporting Bias |
|---|---|---|---|---|---|
| Story et al. 2019 [6] | Low | High | Low | High | Low |
| Burzynski et al. 2020 [9] | Low | High | Low | Low | Low |
| Guo et al. 2020 [7] | Low | High | Low | Low | Unclear |
| Ravenscroft et al. 2020 [8] | Low | High | Unclear | Low | Unclear |

across all studies. Detection bias was low in the studies by Story et al. (2019) [6], Burzynski et al. (2020) [9], and Guo et al. (2020) [7], while Ravenscroft et al. (2020) [8] presented an unclear risk. Attrition bias generally showed a low risk except in Story et al. (2019), which had a high risk. Reporting bias was unclear in half of the studies, indicating variability in how outcomes were reported [6,7,8,9].

## 4 Discussion

In this systematic review of 4 studies and 1044 patients, we compared VOT to DOT in terms of treatment adherence, cost of treatment observation, time spent observing treatment and patient satisfaction. The main findings were as follows: (1) in all RCTs, VOT demonstrated non-inferiority in terms of treatment adherence compared to traditional DOT; (2) VOT reduced costs in the RCTs where this outcome was assessed; (3) the use of VOT reduced the amount of time healthcare professionals spent supervising treatment in the RCTs evaluating this aspect; (4) VOT contributed to higher treatment satisfaction in RCTs where this outcome was measured.

VOT increased the number of completed doses and treatment completion among the participants [6,7,9]. Furthermore, VOT decreased the number of days of missed observation [8]. While VOT offers the advantage of remote supervision, potentially reducing logistical barriers, its efficacy may be influenced by factors such as access to technology and internet connectivity, which might be a problem, particularly in low and middle income countries, where tuberculosis incidence is higher.

Our data indicates that VOT presents a promising alternative for potentially mitigating the financial burden associated with tuberculosis treatment supervision. By facilitating remote observation of medication intake, VOT has the potential to alleviate logistical constraints and minimize the need for in-person healthcare visits, consequently reducing transportation costs and lost productivity for both patients and healthcare providers [6,7,8]. Additionally, VOT interventions offer the possibility of cost savings at a broader health system level. However, it is imperative to acknowledge that the implementation of VOT necessitates initial investment in technological infrastructure and training for healthcare personnel.

The aspect of patient satisfaction merits attention. Our analysis suggests that VOT offers several advantages that may contribute to enhanced patient satisfaction levels. The convenience and flexibility afforded by VOT, allowing patients to take their medication remotely via video calls, can alleviate the burden of frequent clinic visits and disruptions to daily life, potentially impacting satisfaction levels. However, it is crucial to acknowledge that individual preferences and contextual factors play a significant role in shaping patient satisfaction, and further research is needed to explore these dynamics comprehensively [6,7,8,9].

## 5. Study limitations

Our study has several limitations. Firstly, the fact that both patients and treatment observers were not blinded to the intervention introduces potential bias. This lack of blinding may have led to performance bias, which could have significantly influenced the behavior of both patients and observers. For instance, patients aware that they were being observed via video might have been more adherent to their treatment regimen than they would have been under unobserved conditions, thus inflating adherence rates. Additionally, this performance bias could impact the interpretation of data related to secondary outcomes, such as patient satisfaction and time spent observing treatment, as the presence of the video interface might have altered the natural dynamics of treatment observation compared to in-person methods. Therefore, the interpretation of the effectiveness of VOT must be approached with caution, taking

into account the possibility that the unblinded nature of these studies may have led to overestimated benefits. Moreover, the exclusion of drug-resistant tuberculosis patients and the narrow inclusion of elderly patients from all studies may limit the applicability of the findings to only a subset of tuberculosis cases.

It is important to highlight that subgroup statistical analysis of sociodemographic and social determinants, such as age, sex, comorbidities, educational level, history of homelessness and drug or alcohol abuse were not reported by any of the studies included. The absence of this data is significant, as these factors can greatly influence patient adherence and treatment outcomes. For instance, comorbidities and substance abuse may affect both the effectiveness of the treatment and the likelihood of patients adhering to the treatment protocols. The nonexistence of subgroup analysis limits our understanding of how these factors might contribute to disparities in adherence and treatment success. Without this information, it is challenging to identify whether certain groups are disproportionately affected by the interventions studied or if specific barriers are hindering their adherence. This gap in the data could impact the applicability of the study's findings across different patient populations and settings [10].

Additionally, VOT may not be universally applicable due to disparities in internet and technology access. This raises concerns about the generalizability of the findings, particularly to areas with limited internet access [11], more likely to be in low- and middle-income countries (LMICs), which hold the greatest burden of both TB and drug-resistant TB globally [12]. Specific challenges include inadequate internet infrastructure, high costs of internet access, and limited availability of smartphones or devices capable of supporting video calls. Besides that, there may be limited digital literacy among patients and healthcare workers, which can further complicate the implementation of VOT. The lack of reliable technology infrastructure and digital support could significantly hinder the adoption of VOT. Consequently, the effectiveness and feasibility of VOT in these settings might be compromised, making it challenging to apply the study's results broadly. However, as internet and smartphone access continue to improve, particularly in LMICs, VOT may become a more viable option for tuberculosis treatment in these regions [13].

Furthermore, our review's comprehensiveness may have been impacted by the exclusion of non-English studies and the omission of other relevant databases from our search. This limitation potentially restricts the inclusion of valuable data from a broader range of sources. Additionally, publication bias could have influenced our results, as studies with positive findings on VOT are more likely to be published and included in the review, which can lead to an overrepresentation of positive results in the literature. Such selective publication practices could distort the true benefits and overall impact of VOT, thereby affecting the conclusions drawn from our analysis.

## 6. Conclusions

The results of this systematic review, which included over 1,000 patients, suggest that VOT, compared to traditional DOT, enhances treatment adherence, reduces the burden of treatment observation in terms of cost and time, and increases patient satisfaction. Nevertheless, these findings should be interpreted with caution due to some limitations. These include the lack of blinding in the included studies, the exclusion of drug-resistant tuberculosis and elderly patients, the absence of statistical analysis for subgroups based on sociodemographic characteristics and social determinants, and most significantly, the potential impact of access to technology and the internet on the generalizability of the results, as such access may not be available to all populations. Therefore, further research is warranted to ensure VOT's effective integration into diverse healthcare settings, particularly in low- and middle-income countries (LMICs), where its impact could be substantial.

## Supporting information

**S1 PRISMA Checklist. PRISMA 2020 checklist.**
(DOCX)

## Author Contributions

**Conceptualization:** Thiago Areas Lisboa Netto.

**Project administration:** Thiago Areas Lisboa Netto.

**Supervision:** Thiago Areas Lisboa Netto, Taniela Marli Bes.

**Visualization:** Thiago Areas Lisboa Netto.

**Writing – original draft:** Thiago Areas Lisboa Netto, Bruna Dellatorre Diniz, Peter Odutola, Clara Rocha Dantas, Maria Carolina Fonseca Loureiro Caldeira de Freitas, Philip Michael Hefford, Taniela Marli Bes.

**Writing – review & editing:** Thiago Areas Lisboa Netto, Bruna Dellatorre Diniz, Peter Odutola, Clara Rocha Dantas, Maria Carolina Fonseca Loureiro Caldeira de Freitas, Philip Michael Hefford, Taniela Marli Bes.

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
