## [Decision Letter · Decision Letter 0]

29 Jul 2024

Dear Mr. Areas Lisboa Netto,

Thank you very much for submitting your manuscript "Video-observed therapy (VOT) vs directly observed therapy (DOT) for tuberculosis treatment: a systematic review on adherence, cost of treatment observation, time spent observing treatment and patient satisfaction" for consideration at PLOS Neglected Tropical Diseases. As with all papers reviewed by the journal, your manuscript was reviewed by members of the editorial board and by several independent reviewers. The reviewers appreciated the attention to an important topic. Based on the reviews, we are likely to accept this manuscript for publication, providing that you modify the manuscript according to the review recommendations. 

Sincerely,

Husain Poonawala

Academic Editor

Georgios Pappas

Section Editor

Reviewer's Responses to Questions

**Key Review Criteria Required for Acceptance?**

**Methods**

-Are the objectives of the study clearly articulated with a clear testable hypothesis stated?

-Is the study design appropriate to address the stated objectives?

-Is the population clearly described and appropriate for the hypothesis being tested?

-Is the sample size sufficient to ensure adequate power to address the hypothesis being tested?

-Were correct statistical analysis used to support conclusions?

-Are there concerns about ethical or regulatory requirements being met?

Reviewer #1: 1. On l. 93, the authors mentioned that the study was conducted in accordance with PRISMA. A filled-out PRISMA checklist should be available, and the authors should ensure all relevant requirements are fulfilled.

2. On l. 131, the authors should include a citation for the Cochrane risk-of-bias tool.

Reviewer #2: The methods are simple but meet the stated objectives. It would be useful to further define DOT - for example in some studies DOT was in-person delivered to the home, while in others DOT meant that people had to go somewhere for their medication. This difference likely had a substantial effect on adherence and preference. It would also be good to have some descriptors of the people included in these studies, age and others that are included across the studies if possible.

**Results**

-Does the analysis presented match the analysis plan?

-Are the results clearly and completely presented?

-Are the figures (Tables, Images) of sufficient quality for clarity?

Reviewer #1: Results: The current ‘Results’ section could benefit from the following revisions:

1. On ll. 139-140, the authors mention the total number of included studies and participants. Before this, it would be appropriate to describe the results of the search and selection process, from the number of records identified in the search to the number of studies included in the review, and refer to Figure 1.

2. Table 1 could benefit from including relevant demographic characteristics for the included participants such as summary statistics for age, proportion of male/female participants, etc.

3. Ll. 148-151 are perhaps a bit too unspecific. What was the proportion of male/female participants, what were the median or mean ages, what were the specific age ranges?

4. While Table 3 provides a good overview of the quality assessment, a summary of its content should be included in the results section, referring to the table.

Furthermore, in the Discussion section, "Study limitations" could benefit from revision:

1. On ll. 211-214, the authors discuss the implications of treatment observers not being blinded. This part of the discussion could also elaborate on how this potential source of bias might have affected the findings of the review.

2. The limitations of the review process are not currently discussed. It is important to discuss limitations such as the potential impact of excluding non-English studies or the fact that large databases such as Scopus and Web of Science were omitted from the search.

3. It would also be beneficial to discuss the role of publication bias and how it may have affected the results of the review.

Reviewer #2: Results are clear.

**Conclusions**

-Are the conclusions supported by the data presented?

-Are the limitations of analysis clearly described?

-Do the authors discuss how these data can be helpful to advance our understanding of the topic under study?

-Is public health relevance addressed?

Reviewer #1: The conclusion is well-written and offers a concise summary of the main findings and their implications. While it does mention that furhter research is warranted, it should also include some of the limitations from the discussion.

Reviewer #2: One wonders if these studies would be generalizable: for example VOT requires that people have a device with internet access. Populations in this study are a select group and this might be included in the discussion.

**Editorial and Data Presentation Modifications?**

Reviewer #1: 1. In Table 2 in the column "Adherence" on the second row, the text is positioned too far to the left. 

2. Neither Figure 1 nor Table 3 are mentioned in the article. Their content should be summarised under results.

Reviewer #2: (No Response)

**Summary and General Comments**

Reviewer #1: In the paper by Netto et al., the authors systematically reviewed randomized controlled trials (RCTs) comparing video-observed therapy (VOT) to directly observed therapy (DOT). They identified 4 eligible RCTs comprising 1,044 patients. The main findings indicate that all 4 RCTs demonstrated non-inferior or increased adherence for VOT compared with DOT, that VOT reduces costs and time spent by healthcare professionals on supervising patients, and that VOT contributes to higher patient satisfaction. Although not including a meta-analysis, the manuscript convincingly demonstrates how VOT is consistently found to be advantageous compared with DOT in the identified RCTs. The manuscript is generally well-written; however, some revisions are necessary to enhance its clarity and comprehensiveness. In particular, the authors should strongly consider including a filled-out PRISMA checklist and ensure that the review is structured accordingly, which is crucial to ensure transparency of the methodology.

Reviewer #2: This is a very simple analysis, but it is clear and meets its objectives. It would be useful to get down into some of the details of the studies, which are important, as described above.

PLOS authors have the option to publish the peer review history of their article (what does this mean?). If published, this will include your full peer review and any attached files.

Reviewer #1: No

Reviewer #2: No

Figure Files:

Data Requirements:

Reproducibility:

References

---

## [Editor Report · Decision Letter 1]

23 Sep 2024

Dear Mr. Areas Lisboa Netto,

We are pleased to inform you that your manuscript 'Video-observed therapy (VOT) vs directly observed therapy (DOT) for tuberculosis treatment: a systematic review on adherence, cost of treatment observation, time spent observing treatment and patient satisfaction' has been provisionally accepted for publication in PLOS Neglected Tropical Diseases.

Best regards,

Georgios Pappas

Section Editor

Georgios Pappas

Section Editor

Satisfied by the responses

---

## [Editor Report · Acceptance letter]

9 Oct 2024

Dear Mr. Areas Lisboa Netto,

We are delighted to inform you that your manuscript, "Video-observed therapy (VOT) vs directly observed therapy (DOT) for tuberculosis treatment: a systematic review on adherence, cost of treatment observation, time spent observing treatment and patient satisfaction," has been formally accepted for publication in PLOS Neglected Tropical Diseases.

Best regards,

Shaden Kamhawi

co-Editor-in-Chief

Paul Brindley

co-Editor-in-Chief
